# Computerized decision support is an effective approach to select memory clinic patients for amyloid-PET

**Hanneke F. M. Rhodius-Meester**[1,2,3,4]*, **Ingrid S. van Maurik**[1,2,5,6], **Lyduine E. Collij**[7], **Aniek M. van Gils**[1,2], **Juha Koikkalainen**[8], **Antti Tolonen**[8], **Yolande A. L. Pijnenburg**[1,2], **Johannes Berkhof**[5,6], **Frederik Barkhof**[7,9], **Elsmarieke van de Giessen**[1,2,7], **Jyrki Lötjönen**[8], **Wiesje M. van der Flier**[1,2,5,6]

1 Alzheimer Center Amsterdam, Neurology, Amsterdam UMC Location VUmc, Vrije Universiteit Amsterdam, Amsterdam, The Netherlands, 2 Amsterdam Neuroscience, Neurodegeneration, Amsterdam, The Netherlands, 3 Department of Internal Medicine, Geriatric Medicine Section, Amsterdam UMC, Vrije Universiteit Amsterdam, Amsterdam, The Netherlands, 4 Department of Geriatric Medicine, The Memory Clinic, Oslo University Hospital, Oslo, Norway, 5 Epidemiology and Data Science, Amsterdam UMC Location Vrije Universiteit Amsterdam, Amsterdam, The Netherlands, 6 Amsterdam Public Health, Methodology, Amsterdam, The Netherlands, 7 Department of Radiology and Nuclear Medicine, Amsterdam UMC, Vrije Universiteit Amsterdam, Amsterdam, The Netherlands, 8 Combinostics Ltd., Tampere, Finland, 9 Queen Square Institute of Neurology and Centre for Medical Image Computing, University College London, London, United Kingdom

* h.rhodius@amsterdamumc.nl

## Abstract

### Background

The use of amyloid-PET in dementia workup is upcoming. At the same time, amyloid-PET is costly and limitedly available. While the appropriate use criteria (AUC) aim for optimal use of amyloid-PET, their limited sensitivity hinders the translation to clinical practice. Therefore, there is a need for tools that guide selection of patients for whom amyloid-PET has the most clinical utility. We aimed to develop a computerized decision support approach to select patients for amyloid-PET.

### Methods

We included 286 subjects (135 controls, 108 Alzheimer's disease dementia, 33 frontotemporal lobe dementia, and 10 vascular dementia) from the Amsterdam Dementia Cohort, with available neuropsychology, APOE, MRI and [18F]florbetaben amyloid-PET. In our computerized decision support approach, using supervised machine learning based on the DSI classifier, we first classified the subjects using only neuropsychology, APOE, and quantified MRI. Then, for subjects with uncertain classification (probability of correct class (PCC) < 0.75) we enriched classification by adding (hypothetical) amyloid positive (AD-like) and negative (normal) PET visual read results and assessed whether the diagnosis became more certain in at least one scenario (PCC≥0.75). If this was the case, the actual visual read result was used in the final classification. We compared the proportion of PET scans and patients diagnosed with sufficient certainty in the computerized approach with three scenarios: 1)

**Data Availability Statement:** The data from this study are available upon request. Subjects included in this study only consented to their data being shared with third parties when it is used to gain

more knowledge about Alzheimer's disease and dementia. Therefore, the data may only be shared via a data-request procedure. The data underlying the results presented in this study are therefore available upon request from the Alzheimer Center Amsterdam, via wm.vdflier@amsterdamumc.nl or metc@vumc.nl. More information on the Cohort can be found at JPND, Amsterdam dementia cohort - JPND Neurodegenerative Disease Research (neurodegenerationresearch.eu).

Funding: The Vrije Universiteit Medical Center Alzheimer Center is supported by the Stichting Alzheimer Nederland and Stichting Vrije Universiteit Medical Center Fonds. The clinical database structure was developed with funding from Stichting Dioraphte. For development of the PredictAD tool, VTT Technical Research Centre of Finland has received funding from European Union's Seventh Framework Programme for research, technological development and demonstration under grant agreements 601055 (VPH-DARE@IT) ), 224328 (PredictAD), and 611005 (PredictND). The collaboration project DAILY (project number LSHM19123-HSGF) is co-funded by the PPP Allowance made available by Health-Holland, Top Sector Life Sciences & Health, to stimulate public-private partnerships. The ABIDE clinical utility study is funded by the PPP Allowance made available by Health-Holland, Top Sector Life Sciences & Health, to stimulate public-private partnerships and co-funded by Life Molecular Imaging GmbH (grant no. LSHM18075). HR is the recipient of the Memorable Dementia Fellowship 2021 (ZonMw project number 105510022110004) and Alzheimer Nederland InterACT grant (project number WE.08-2022-06). LC is supported by the Innovative Medicines Initiative 2 Joint Undertaking under grant agreement no. 115952. This Joint Undertaking receives the support from the European Union's Horizon 2020 research and innovation program and EFPIA. FB is supported by the NIHR biomedical research centre at UCLH. The chair of WF is supported by the Pasman Stichting. WF is recipient of ABOARD, which is a public–private partnership receiving funding from ZonMW (number 73305095007) and Health-Holland, Top Sector Life Sciences and Health (public–private partnership allowance; number LSHM2010). HR and WF are recipients of Prominent. The Prominent project is supported by the Innovative Health Initiative Joint Undertaking (JU) under grant agreement no. 101112145. The JU receives support from the European Union's Horizon Europe research and innovation programme and COCIR, EFPIA, EuropaBio MedTech Europe, Vaccines Europe, BioArctic AB and Combinostics Oy. Views and opinions expressed are those of the authors

without amyloid-PET, 2) amyloid-PET according to the AUC, and 3) amyloid-PET for all patients.

## Results

The computerized approach advised PET in n = 60(21%) patients, leading to a diagnosis with sufficient certainty in n = 188(66%) patients. This approach was more efficient than the other three scenarios: 1) without amyloid-PET, diagnostic classification was obtained in n = 155(54%), 2) applying the AUC resulted in amyloid-PET in n = 113(40%) and diagnostic classification in n = 156(55%), and 3) performing amyloid-PET in all resulted in diagnostic classification in n = 154(54%).

## Conclusion

Our computerized data-driven approach selected 21% of memory clinic patients for amyloid-PET, without compromising diagnostic performance. Our work contributes to a cost-effective implementation and could support clinicians in making a balanced decision in ordering additional amyloid PET during the dementia workup.

## Introduction

The neuropathological hallmark of Alzheimer's disease (AD), amyloid-beta, can be visualized via amyloid positron emission tomography (PET) [1–3]. After having shown clinical impact in memory clinic patients, the use of amyloid-PET in daily clinical practice is upcoming, both for accurate and etiological diagnosis and to initiate disease-modifying treatment (DMT) [4–8]. At the same time, amyloid-PET is costly and limitedly available outside tertiary memory clinics. There is a need for tools that can aid clinicians in identifying which patient would benefit from amyloid-PET to ensure an accurate etiological diagnosis, whilst remaining efficient [9–12].

The Amyloid Imaging Task Force (AIT) has developed appropriate use criteria (AUC) based on expert opinion, to foster the optimal use of amyloid-PET [13]. Amyloid-PET is deemed appropriate in patients with possible AD 'for whom substantial uncertainty exists and for whom greater confidence would result from determining whether amyloid pathology is present or not'. Also, amyloid-PET may be performed in young-onset dementia to increase diagnostic confidence [13]. Despite the efforts of the AIT, the AUC are not sufficiently able to discriminate between patients who would benefit from amyloid-PET and those who would not [14–16]. For example, we showed that in an unselected memory clinic cohort, patients not fulfilling the AUC also benefited from amyloid-PET [14]. Translation of the AUC to clinical practice is thus challenging, hampering successful implementation of this expensive test in memory clinicians [17]. Studies have repeatedly shown that amyloid-PET increases diagnostic confidence. Nonetheless, it is likely that in some patients diagnostic confidence was already high enough before amyloid-PET, whilst in others confidence may remain low, even after amyloid PET. Knowledge on how amyloid status would impact etiological diagnosis in individual patients would help clinicians to decide which patient should undergo an amyloid-PET and which patient should not.

We previously developed a computerized decision support approach to support clinicians in identifying patients most likely to benefit from cerebrospinal fluid (CSF) biomarkers [18].

and do not necessarily reflect those of the aforementioned parties. Neither of the aforementioned parties can be held responsible for them.

**Competing interests:** We have read the journal's policy and the authors of this paper have the following competing interests: Hanneke F.M. Rhodius-Meester performs contract research for Combinostics; all funding is paid to her institution. Ingrid van Maurik received a consultancy fee (paid to the university) from Roche. Lyduine E. Collij has received consultancy fees from GE Healthcare; all funding is paid to her institution. Juha Koikkalainen and Jyrki Lötjönen report that Combinostics Oy owns the following IPR related to the paper: 1. J. Koikkalainen and J. Lotjonen, "A method for inferring the state of a system," US7,840,510 B2, PCT/FI2007/050277. 2. J. Lotjonen, J. Koikkalainen and J. Mattila, "State Inference in a heterogeneous system," PCT/FI2010/050545, FI20125177. Koikkalainen and Lötjönen are shareholders in Combinostics Oy. Yolande A.L. Pijnenburg has received funding from Dioraphte Foundation, Zabawas Foundation, JPND, ZonMW, NWO, Team Alzheimer and the Dutch Brain Foundation. Frederik Barkhof is member of the Steering committee or Data Safety Monitoring Board member for Biogen, Merck, ATRI/ACTC and Prothena. Barkhof is a consultant for Roche, Celltrion, Rewind Therapeutics, Merck, IXICO, Jansen, Combinostics. Barkhof has research agreements with Merck, Biogen, GE Healthcare, Roche. Co-founder and shareholder of Queen Square Analytics LTD. Elsmarieke van de Giessen has received research support from NWO, ZonMw, Hersenstichting and KWF. van de Giessen has performed contract research for Heuron Inc., Roche and 1st Biotherapeutics. van deGiessen has a consultancy agreement with IXICO for the reading of PET scans. Wiesje M van der Flier performs contract research for Biogen. Research programs of van der Flier have been funded by ZonMW, NWO, EU-FP7, EU-JPND, Alzheimer Nederland, CardioVascular Onderzoek Nederland, Health~Holland, Topsector Life Sciences & Health, stichting Dioraphte, Gieskes-Strijbis fonds, stichting Equilibrio, Pasman stichting, stichting Alzheimer & NeuroPsychiatry Foundation, Philips, Biogen MA Inc, Novartis-NL, Life-MI, AVID, Roche BV, Fujifilm, Combinostics. van der Flier has performed contract research for Biogen MA Inc, and Boehringer Ingelheim. van der Flier has been an invited speaker at Boehringer Ingelheim, Biogen MA Inc, Danone, Eisai, WebMD Neurology (Medscape), Springer Healthcare. van der Flier is consultant to Oxford Health Policy Forum CIC, Roche and Biogen MA Inc. van der Flier participated in advisory boards of Biogen MA Inc.

This data-driven approach restricted CSF testing to 26% of cases without compromising diagnostic accuracy. In this work, we took a similar data-driven approach to predict which patients would benefit from amyloid-PET testing. More specifically, we tested whether this approach may help to answer the following question: if a clinician already has detailed information on neuropsychological tests, APOE and brain imaging, would additional amyloid-PET contribute to a more certain etiological diagnosis?

## Material and methods

### Subjects

We retrospectively included 286 memory clinic subjects who visited our memory clinic seeking medical help between January 2015 and December 2016 (the Amsterdam Dementia Cohort) with a diagnosis of Alzheimer's dementia (AD), frontotemporal lobe dementia (FTD), vascular dementia (VaD), or subjective cognitive decline (SCD) [19, 20]. As part of the ABIDE (Alzheimer Biomarkers in Daily practice) project [5, 21], [18F]florbetaben PET was offered for clinical care to all consecutive memory clinic patients between January 2015 and December 2016. Subjects who had both amyloid-PET and brain MRI results available were included.

All subjects received a standardized work-up at baseline to come to a diagnosis, including medical history, physical, neurological and neuropsychological assessment, MRI, laboratory tests, and amyloid-PET. A diagnosis of SCD was made when the cognitive complaints could not be confirmed by cognitive testing and criteria for mild cognitive impairment (MCI) or dementia were not met. Subjects with SCD served as controls. Probable AD was diagnosed using the core clinical criteria of the NIA-AA [22]. Probable FTD (including the behavioural variant of FTD, progressive non-fluent aphasia, and semantic dementia) was diagnosed using the criteria from Rasckovsky and Gorno-Tempini, respectively [23, 24]. VaD was diagnosed using the NINDS-AIREN criteria [25].Since the classifier we used for this study (see below for detailed description) is currently only able to classify controls, patients with AD, FTD and VAD, subjects with other diagnoses, such as dementia with Lewy bodies (DLB) were not included.

The data in this study were collected during routine care and retrieved retrospectively. The Daily Board of the Medical Ethical Committee (METc) of the VUmc Medical Center provided an exemption to seek formal approval. All patients provided written informed consent for their data to be used for research purposes. The authors had no access to information that could identify individual participants during or after data collection.

### Neuropsychology testing

Cognitive functions were assessed with a brief standardized test battery, including widely used tests. We used the Mini-Mental State Examination (MMSE) for global cognitive functioning [26]. For memory, we applied the Rey auditory verbal learning task (RAVLT) [26]. To measure mental speed and executive functioning, we included Trail Making Tests A and B (TMT-A, TMT-B) [27]. Language and executive functioning were tested by category fluency (animals) [28]. Finally, for behavioral symptoms, we used the Neuropsychiatric Inventory (NPI) [29]. Missing data ranged from n = 3 (1%) (MMSE) to n = 75 (26%) (NPI).

### APOE genotype

Apolipoprotein E (APOE) genotype was determined with the light cycler APOE mutation detection method (Roche diagnostics GmbH, Mannheim, Germany). Patients were

and Roche. All funding is paid to her institution. van der Flier was associate editor of Alzheimer, Research & Therapy in 2020/2021. van der Flier is associate editor at Brain. This does not alter our adherence to PLOS ONE policies on sharing data and materials.

dichotomized into APOE e4 carriers (hetero- and homozygous) and non-carriers. APOE data were available in 283 (99%) subjects.

## Imaging markers

MRI images were acquired using 1.5 T or 3 T scanners including 3D isotropic T1 and 2D or 3D FLAIR sequences. We extracted six imaging markers using the cNeuro® cMRI quantification tool as described in [18]:

- Computed medial temporal lobe atrophy (cMTA) was computed for the left and right hemispheres from the volumes of the hippocampus and inferior lateral ventricle as described in [30, 31]. The volumes were obtained from a multi-atlas segmentation algorithm [32].

- Computed global cortical atrophy (cGCA) measured the gray matter concentration based on the voxel-based morphometry (VBM) analysis [30, 31].

- AD similarity scale was computed by representing the patient image as a linear combination of regional volumes from a database of previously diagnosed patients [17, 38]. The AD similarity scale was defined as the share of the weights from the linear model having the diagnostic label AD.

- Anterior-posterior index was defined as a ratio of the cortical volumes at frontal and temporal lobe regions to those at parietal and occipital lobe regions [33].

- The volume of white matter hyperintensities (WMH) was extracted from FLAIR images [31, 34].

## Amyloid-PET

Procedures for amyloid-PET using [18F]florbetaben have been described in detail elsewhere [5, 21]. Per standard protocol, 20-minute scans consisting of 4x5 minute frames were collected 90–110 minutes post-injection of approximately 300 MBq±20% [18F]florbetaben (Neuraceq™, Life Molecular Imaging, Berlin, Germany). We used visual reads and repeated the analyses using Centiloids. Visual reads were available in all subjects, Centiloids in 248 (87%).

PET scans were visually assessed by a certified and experienced nuclear physician blinded to clinical diagnosis. Images were scaled based on the total white matter signal and grey color scaling. Transverse, sagittal, and coronal views were displayed using the software package Vinci 2.56. Images were rated as either *positive* (binding in one or more cortical brain regions unilaterally) or *negative* (predominantly white matter uptake) according to criteria defined in the label by the manufacturer (Life Molecular Imaging).

For Centiloid quantification, all scans were pre-processed using a validated standard Centiloid pipeline and converted to the Centiloid scale[34]. Briefly, the four frames from the PET images were first averaged and co-registered to the corresponding T1-weighted scans. Then, the T1- weighted MRI scans were warped to standard space; the same warp was applied to warp the co-registered PET image. These procedures were performed in SPM12. PET images were intensity normalized using the whole cerebellum as the reference region using the mask provided by the Centiloid method [34] (http://www.gaain.org/centiloid-project). Global cortical Centiloid values were calculated using the standard GAAIN target region Centiloid. Centiloid calibration has been previously described [35].

## Disease State Index classifier and probability of correct class

The Disease State Index (DSI) classifier was previously developed and validated in the European FP7 PredictND project [36, 37]. The DSI is a simple, supervised, and data-driven machine learning method that compares different diagnostic groups with each other; in this work controls, AD, FTD, and VaD. There is no need to impute data or exclude cases with incomplete data, as the classifier can handle missing data. The classifier is based on a training set with diagnosed patients [36]. For each single test (e.g. neuropsychological, APOE-status, MRI), the similarities of each patient's data to the distributions of the diagnostic groups in the training set are computed. When single tests are combined, the tests with higher classification accuracy are weighted more. First, a DSI value is calculated for each pair-wise comparison (AD-controls, AD-FTD, AD-VaD, FTD-controls, FTD-VaD, VaD-controls). Then, the final DSI-value for each diagnostic group (controls-AD-FTD-VaD) is calculated by averaging the corresponding DSI-values, e.g AD-controls, AD-FTD and AD-VaD for AD etc, as described in [18]. As a result, a DSI value (continuous value between zero and one) is given for each diagnostic group (controls-AD-FTD-VaD), estimating the likelihood of the specific diagnosis.

This study was performed using five-fold cross-validation, i.e., 80% of the dataset was used as the training set in the DSI classifier when classifying the remaining 20% of the patients. This was repeated five times so that each patient was classified once.

A high DSI value or a big difference in DSI values between the two most likely diagnostic groups provides more certainty in making a diagnosis than a low value or a small difference [18]. The probability that the diagnostic group of the highest DSI value is correct is defined as the probability of correct class (PCC). The diagnosis suggested is compared with the ground truth diagnosis for the cases with comparable highest DSI value and the difference between two highest DSI values in a reference database and the share of correct diagnoses is calculated. The reference database used consisted of 770 memory clinic patients (Amsterdam dementia cohort and PredictND) diagnosed with the same guidelines as used in the current study [20, 37]. That dataset consisted of 308 controls, 338 ADs, 89 FTDs, and 35 VaDs. The mean age was 65.8 ± 8.7 years, and 54% were females.

In this study, patients were considered as having a diagnosis with sufficient certainty if PCC was ≥0.75. This selection was a compromise between the number of diagnosed patients and accuracy. In clinical practice, the clinician can adjust the applied PCC cutoff depending on the pre-test probability.

## Diagnostic scenario's to select patients for amyloid-PET

We applied four diagnostic scenarios (Fig 1) in which patients were considered as having a diagnosis with sufficient certainty if the PCC was ≥0.75. As amyloid-PET measures, we used visual reads and repeated our analyses using Centiloid quantification.

- Scenario A: In this *Computer-supported decision approach*, we performed amyloid-PET only when predicted to change the certainty in diagnosis based on our data-driven method (Fig 1A). Patients were regarded as sufficient certain cases if PCC was ≥0.75 based on APOE, neuropsychology, and MRI (step one). When PCC was <0.75, the computer tool added both (hypothetical) positive and negative amyloid PET. Hypothetical PCC's were computed for both positive amyloid-PET and negative amyloid-PET values (step two). If either of these hypothetical PCC values reached >0.75, the actually observed amyloid-PET values were added, after which DSI and PCC, were computed (step three). On repeating this scenario using Centiloid values, we took the mean Centiloid value for AD patients (69.40 ± 39.3) and the mean Centiloid value for controls (11.95 ± 24.31) as hypothetical values in step two.

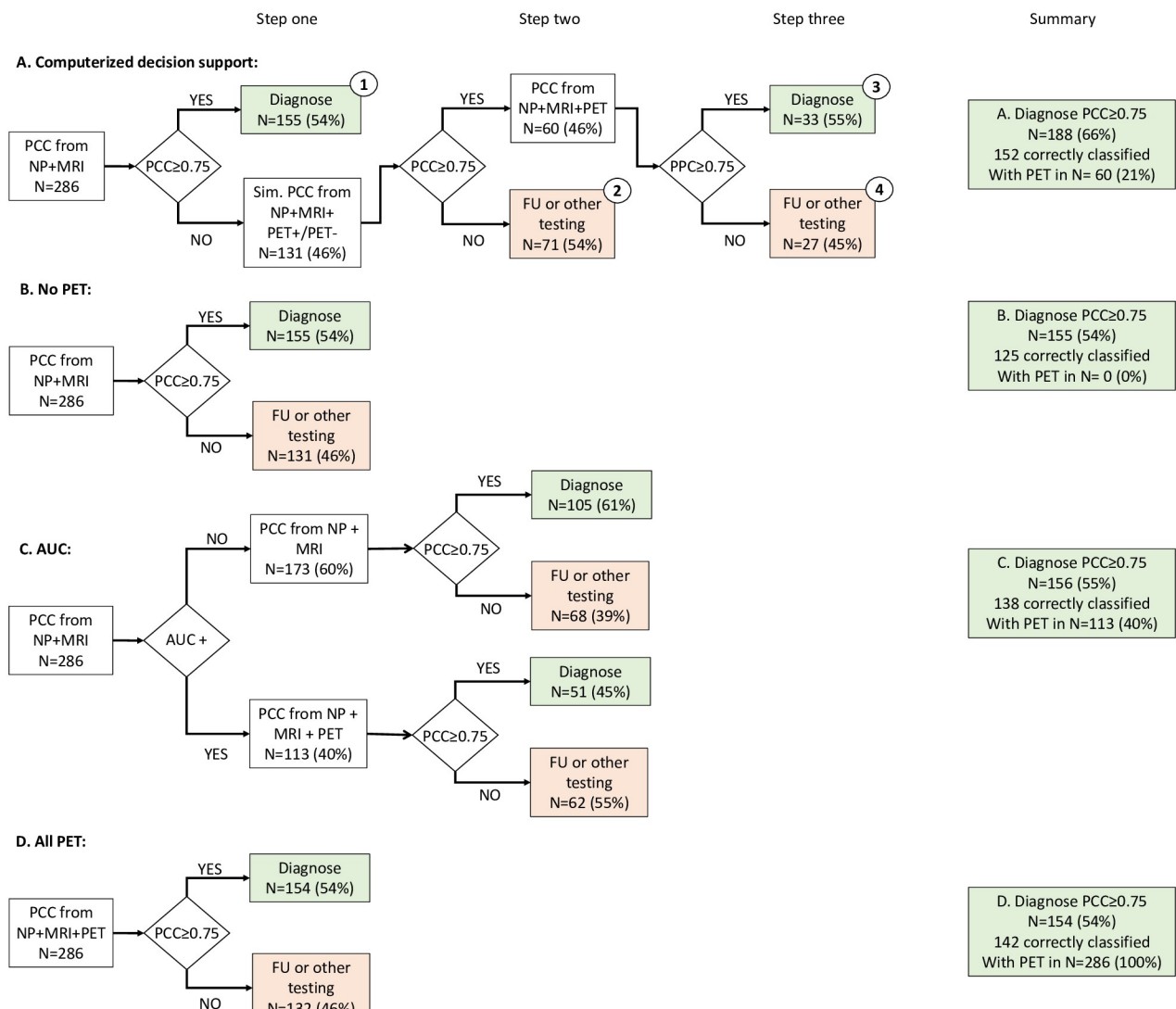

**Fig 1. Flow chart for the four diagnostic approaches, using amyloid-PET visual read, summarizing the results in the last column.** AUC: appropriate use criteria, AUC+: patients fulfilling appropriate use criteria according to [13], operationalized as described in [14], PCC: probability of correct class, NP: neuropsychology, MRI: magnetic resonance imaging, Sim: simulate, FU: follow-up. Numbers in circles denote groups described in Table 2.

We compared this approach, with the following three 'control scenario's:

- Scenario B: In the *No amyloid-PET approach*, we calculated DSI and PPC for each patient using only APOE, neuropsychology and MRI, excluding amyloid-PET (Fig 1B).

- Scenario C: In the *AUC scenario*, we performed amyloid-PET based on the AUC criteria (Fig 1C). We classified patients as AUC-positive (AUC+) and AUC-negative (AUC-), according to [14]. In this paper patients were classified during pre-PET multidisciplinary meetings as AUC+ when they either i) had AD as diagnostic possibility ($\geq$15%) but with a confidence <85% in AD as diagnosis, or ii) had a young-onset dementia (<65 years old. All other patients were classified as AUC-[38]. In patients classified AUC+, we calculated PCC by adding amyloid-PET (step two). For AUC- patients, no amyloid-PET was added.

- Scenario D: In the *All amyloid-PET approach* (Fig 1D)., we calculated DSI and PCC for each patient using APOE, neuropsychology, MRI, and amyloid-PET.

For all four approaches, we reported the number of patients diagnosed with sufficient certainty and the number of patients in which PET was performed.

## Statistical analyses

Further scrutinizing the data of scenario A, we tested differences in baseline characteristics, diagnosis, and DSI between patients with sufficiently certain diagnosis based directly on neuropsychology and MRI (step one in computerized decision support approach, group 1 in Fig 1A), patients not eligible for amyloid-PET testing (step two, group 2) and patients with actual amyloid-PET testing (step three, groups 3 and 4).

Lastly, we visualized the impact of different PCC cut-offs for the proportion of patients diagnosed (percentage of patients above PCC cutoff) and the proportion of patients with amyloid-PET measurement for all four diagnostic approaches described above.

MRI markers were normalized for age, sex and head size [39]. Statistical analyses were performed using SPSS version 22 (IBM, Armonk, NY, USA), STATA version 14.1, and R version 3.5.3. A MATLAB toolbox created by [40] was used in the DSI analyses. The analyses were performed in MATLAB version R2018b (MathWorks, Natick, MA, USA).

## Results

### Baseline characteristics

In the study sample, the mean age was 64±8 and 129 (45%) were females. Table 1 shows details of the baseline characteristic of this sample, stratified per diagnostic group.

### Diagnostic approaches to select patients for amyloid-PET

In our search for the optimal approach to select patients for amyloid-PET, we applied four diagnostic approaches. Fig 1 shows the flowchart of these four approaches and summarizes the number of patients with sufficient certain diagnoses (PCC≥0.75), and the number of patients selected for amyloid-PET. In these results, the amyloid-PET biomarker was the visual read. First, we applied the computerized decision support approach (scenario A). Using demographics, APOE, neuropsychology and MRI only, diagnostic prediction was sufficiently certain (PCC ≥0.75) in 155 (54%) cases. In the 131 (46%) remaining cases, hypothetical positive and negative amyloid-PET values were added (step 1), and this led to an increase of PCC to ≥0.75 in 60 (46%) cases, thus advising performing an amyloid-PET scan (step 2). When real amyloid-PET values were actually added to the model, we observed a PCC≥0.75 in 33 (55%) patients (step 3). Overall, the computerized approach led to a diagnosis with sufficient confidence in 188 (66%) patients by performing PET in 60 (21%) patients, with correct classification of 152 patients.

We compared our data-driven approach to three control scenario's. In scenario B, the scenario without amyloid- PET, we used demographics, APOE, neuropsychology, and MRI only, and found a diagnosis with sufficient confidence in 155 patients (54%) from which 125 were correctly classified. Scenario C, applying amyloid-PET based on the AUC, led to amyloid-PET in a larger group of 113 (40%) patients, yet not to a higher proportion of patients with a certain diagnosis, 156 (55%), and correctly classifying 138 patients. In scenario D, performing amyloid-PET in all patients, again did not lead to more diagnoses with sufficient confidence, namely 154 (54%), and correct classification in 142 patients.

**Table 1. Baseline characteristics according to baseline diagnosis.**

|  | n = | Control n = 135 | AD n = 108 | FTD n = 33 | VaD n = 10 |
|---|---|---|---|---|---|
| Female, n(%) | 286 | 58 (43) | 56 (52) | 13 (39) | 2 (20) |
| Age, in years | 286 | 60 ± 8 | 66 ± 7 | 66 ± 7 | 72 ± 6 |
| APOE e4 carrier, n(%) | 283 | 53 (39) | 74 (69) | 11 (33) | 2 (20) |
| *Neuropsychology* |  |  |  |  |  |
| MMSE | 283 | 28 ± 2 | 22 ± 4 | 24 ± 5 | 24 ± 4 |
| RAVLT learning | 275 | 39 ± 10 | 22 ± 7 | 26 ± 11 | 23 ± 9 |
| RAVLT recall | 275 | 8 ± 3 | 2 ± 2 | 4 ± 3 | 3 ± 3 |
| Animal fluency | 269 | 22 ± 6 | 14 ± 5 | 11 ± 6 | 12 ± 5 |
| TMT-A, in seconds | 273 | 39 ± 22 | 92 ± 84 | 70 ± 55 | 70 ± 24 |
| TMT-B, in seconds | 273 | 100 ± 65 | 294 ± 252 | 204 ± 171 | 219 ± 69 |
| NPI, total score | 211 | 14 ± 16 | 9 ± 9 | 17 ± 15 | 19 ± 21 |
| *MRI* |  |  |  |  |  |
| cMTA score right | 283 | 0.31 ± 0.51 | 1.37 ± 0.94 | 1.54 ± 1.20 | 1.34 ± 1.10 |
| cMTA score left | 283 | 0.31 ± 0.58 | 1.49 ± 1.07 | 1.86 ± 1.48 | 1.99 ± 1.30 |
| cGCA score | 283 | 0.39 ± 0.58 | 1.52 ± 0.86 | 1.89 ± 0.95 | 2.15 ± 0.81 |
| Anterior Posterior index | 283 | -0.25 ± 1.23 | 0.03 ± 1.76 | -1.84 ± 2.79 | -0.44 ± 1.53 |
| AD similarity scale | 283 | 0.47 ± 0.11 | 0.63 ± 0.09 | 0.57 ± 0.10 | 0.61 ± 0.07 |
| WMH volume | 279 | 6.00 ± 5.33 | 9.90 ± 13.1 | 11.08 ± 14.06 | 21.01 ± 18.34 |
| *Amyloid-PET* |  |  |  |  |  |
| Visual Read (neg/pos) | 286 | 107 / 28 | 5 / 103 | 31 / 2 | 6 / 4 |
| Centiloids; mean ± SD | 248 | 11.95 ± 24.31 | 69.40 ± 39.3 | 7.52 ± 21.28 | 15.93 ± 22.87 |
| AUC +, n(%) | 286 | 6 (4) | 81 (75) | 20 (61) | 6 (60) |

AD: Alzheimer's disease, FTD: Frontotemporal dementia, VAD: Vascular dementia, MMSE: Mini-Mental state Examination, RAVLT: Rey Auditory Verbal Learning Test, TMT: Trail Making Test, NPI: Neuropsychiatric Inventory score, cMTA: computed medial temporal lobe atrophy scale (0–4), derived from volume of hippocampus and volume of inferor lateral ventricle, cGCA: computed global cortical atrophy scale (0–3), derived from concentration of cortical grey matter using voxel based morphometry, AD similarity scale: based on hippocampus ROI, Anterior posterior index: weighted ratio of volumes of the frontal/temporal lobes and parietal/occipital lobes WMH: volume of white matter hyperintensities. MRI volumes are adjusted for head size, AUC +: number of patients fulfilling appropriate use criteria according to [13], operationalized as described in [14].

Using Centiloid values instead of visual reads yielded similar results (see S1 Fig).

## Differences in patients groups using computerized decision support approach

Following the flowchart of the computerized decision support approach in Fig 1A, four distinct groups can be separated in the three steps, marked with 1-2-3-4 in the figure and summarized in Table 2. In the first group are those patients with a diagnosis with sufficient certainty, using only demographics, APOE, neuropsychology, and MRI. This group was the most extensive (n = 155) and contained patients with all types of diagnoses. These patients had the largest difference in DSI value between the first and second suggested diagnoses. Presumably, this group had a clear, distinct profile, both clinically and on imaging, and little co-morbidity. This group contrasts with the second group, containing the patients in which adding hypothetical amyloid-PET values did not increase diagnostic certainty (n = 71). Here, the difference between first and the second DSI was the smallest, indicating co-morbid neuropathology or neuropsychological profiles that are hard to distinguish from each other. This group could not be certainly diagnosed neither with nor without amyloid PET, and follow-up or other testing

**Table 2. Comparison of different patient groups deriving from the computerized decision support approach using visual reads; matching Fig 1A.**

| | 1. Direct sufficient certain diagnosis | 2. PET not useful | 3. PET helpful to establish diagnosis | 4. Not diagnosed |
|---|---|---|---|---|
| | n = 155 | n = 71 | n = 33 | n = 27 |
| Female, n(%) | 68 (44) | 30 (42) | 14 (42) | 17 (63) |
| Age, in years | 62 ± 8 | 64 ± 8 | 67 ± 8 | 66 ± 6 |
| APOE e4 carrier, n(%) | 76 (49) | 33 (46) | 17 (52) | 14 (56) |
| MMSE | 26 ± 4 | 26 ± 3 | 22 ± 5 | 23 ± 4 |
| *Amyloid-PET, visual read* | | | | |
| Negative | 87 (56) | 44 (62) | 12 (36) | 6 (22) |
| Positive | 68 (44) | 27 (38) | 21 (64) | 21 (78) |
| *Clinical diagnosis* | | | | |
| Control | 87 (64) | 41 (30) | 3 (2) | 4 (3) |
| AD | 47 (44) | 20 (19) | 23 (21) | 18 (17) |
| FTD | 17 (52) | 8 (24) | 5 (15) | 3 (9) |
| VaD | 4 (40) | 2 (20) | 2 (20) | 2 (20) |
| Difference with second DSI | 0.33 ± 0.09 | 0.08 ± 0.05 | 0.10 ± 0.06 | 0.11 ± 0.06 |
| AUC +, n(%) | 57 (37) | 24 (34) | 20 (61) | 16 (59) |

MMSE: Mini-Mental state Examination, AD: Alzheimer´s disease, FTD: Frontotemporal dementia, VAD: Vascular dementia, DSI: Disease State Index, Difference with second DSI: DSI based on demographics, APOE, neuropsychology and MRI, AUC +: number of patients fulfilling appropriate use criteria according to [13], operationalized as described in [14]

is advised. The third group included those patients in whom the computerized approach suggested amyloid-PET according to hypothetical +/- amyloid-PET results (n = 33). After adding actual PET values, the PCC increased to ≥0.75. In this group, the amyloid-PET scan was often positive (64%) and contained mainly patients with AD (23/33). The final group consisted of the patients for whom, despite performing amyloid-PET scan, the diagnosis remained unclear (n = 27).

## Effect of different PCC cutoffs on diagnosis and amyloid-PET

For the results described above, we used PCC≥0.75 to define which diagnostic prediction is accurate and has sufficient certainty. Of note, this is an arbitrary choice. To study the effect of the PCC cutoff, we repeated all our analyses for different PCC cutoffs ranging from 0.5 to 1.0. In Fig 2, we compared all four approaches based on the proportion of patients diagnosed with sufficient certainty (Fig 2A) and the number of performed amyloid-PET scans (Fig 2B) using different PCC cutoffs. As expected, the share of patients with certain diagnosis declined with increasing PCC, independent of the scenario used. Overall the proportion of diagnosed patients was largest when using the computerized decision support approach and the lowest when performing no PET, independent of the PCC cutoff.

## Example of visualization of computerized decision support in clinical practice

How the computerized decision support approach could be used in clinical practice is visualized in Fig 3. Case A, for example, is a 65-year-old female, who experiences memory problems, but also scores low on fluency and high on NPI, while MRI showed hardly atrophy. Based on demographics, neuropsychology, and MRI, the classifier suggested an FTD diagnosis (DSI 0.72) yet with a minimal difference to the next most probable diagnosis AD (DSI 0.71).

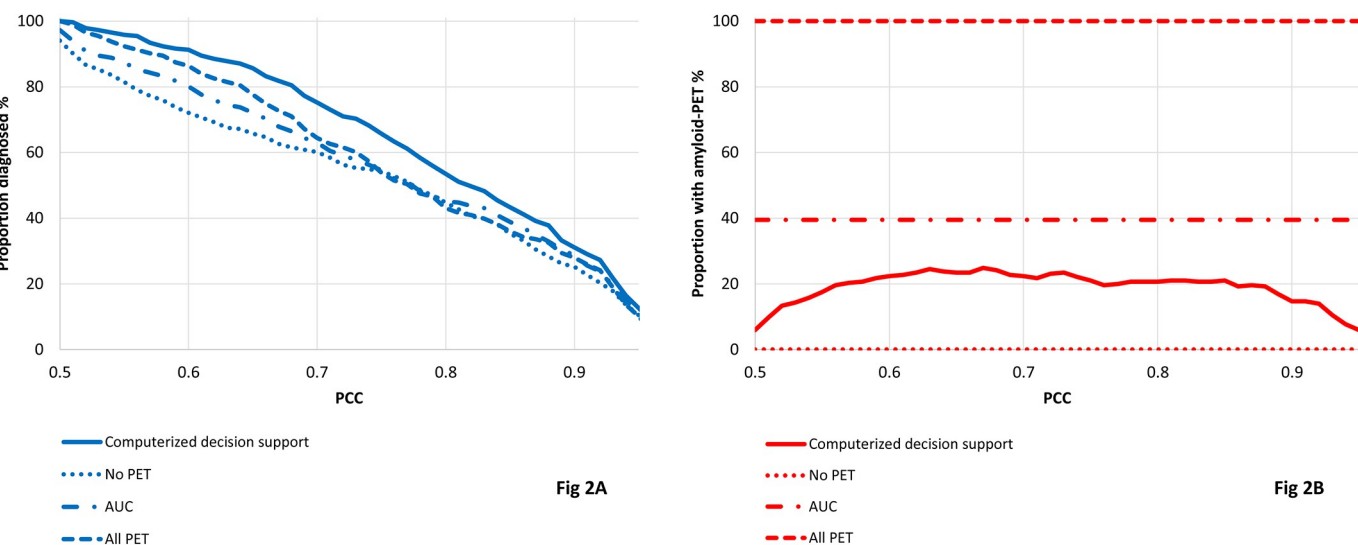

**Fig 2. Visualization of the share of patients diagnosed (blue, 2A) and the share of patients with amyloid-PET performed (red, 2B) for different probability of correct class cutoffs, comparing computerized decision support, no amyloid-PET, AUC, and amyloid-PET for all patients.** Blue: proportion of patients diagnosed, Red: proportion of patients with amyloid-PET taken, PCC: probability of correct class. Solid lines show results for the computerized decision support (Fig 1A), dotted lines show results for using no amyloid-PET, but only demographics, APOE, neuropsychology and MRI (Fig 1B), dashed dotted lines show results for AUC (Fig 1C) and dashed lines using all data (Fig 1D).

Therefore, the probability of correct class (PCC) is low (0.51). When the tool adds hypothetical positive and negative amyloid-PET scans, the clinician can see that both a positive and a negative amyloid-PET result, would influence the diagnostic certainty (PCC > 0.75 in both situations). The lower panel shows results after addition of the actual amyloid PET scan, which was positive in this case, leading to a high PCC (0.78) for AD diagnosis (DSI 0.81). In case B, a 71–year-old female who has trouble performing the cognitive tests due to impaired understanding, yet surprisingly does perform TMT-A relatively fast, whereas MRI showed mild bitemporal atrophy. The classifier showed a low PCC (0.50) using only demographics, neuropsychology, and MRI, with equal DSI for both AD and FTD diagnosis (DSI 0.63). In this case, adding a hypothetical positive or negative amyloid-PET changed the PCC to >0.75 for a negative amyloid PET scan (albeit not for a positive PET scan). Based on an increase to >0.75 in one of the scenario's, the clinician is advised to embark on ordering an amyloid-PET scan, which in this case was negative. A clinical diagnosis of probable FTD was confirmed.

## Discussion

In this study, a data-driven approach in which diagnostics classification is enriched by adding (hypothetical) amyloid positive (AD-like) and negative (normal) PET to aid the clinician in deciding whether performing an actual amyloid-PET scan contributed to a more certain diagnosis. Our computerized decision support approach advised performing an amyloid-PET scan in 21% of the patients without compromising proportion of correctly classified cases. Our approach was thus more efficient than the other scenario's, where we would have performed PET in all patients, in none, or according to the appropriate use criteria (AUC). When implemented in a computer tool, this approach can support clinicians in making a balanced decision in ordering additional (expensive) amyloid-PET testing using personalized patient data.

Approaches such as the data-driven approach we demonstrated in this study, can aid in translating appropriate use criteria (AUC) to clinical practice. The AUC state that amyloid-

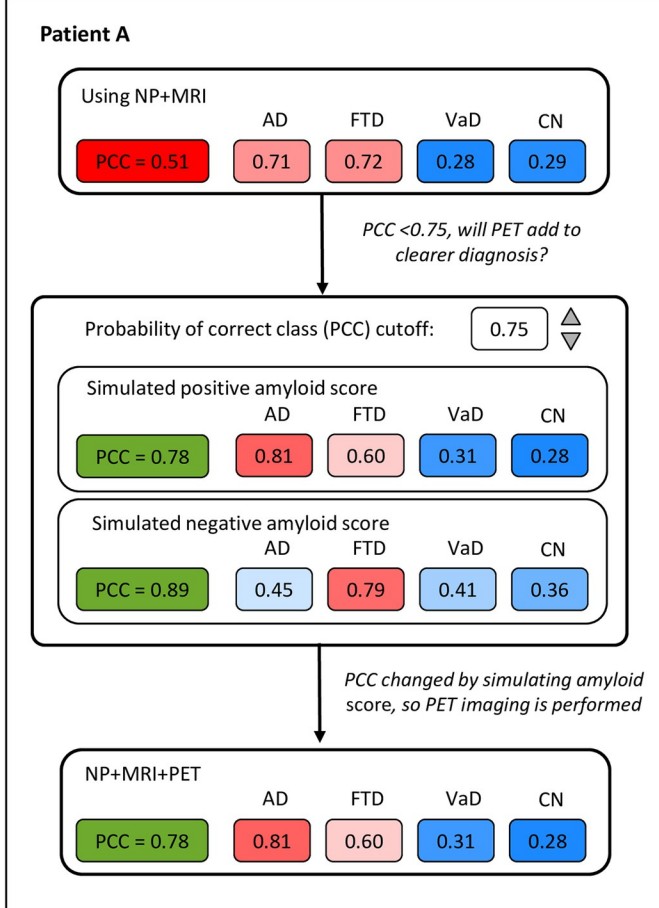
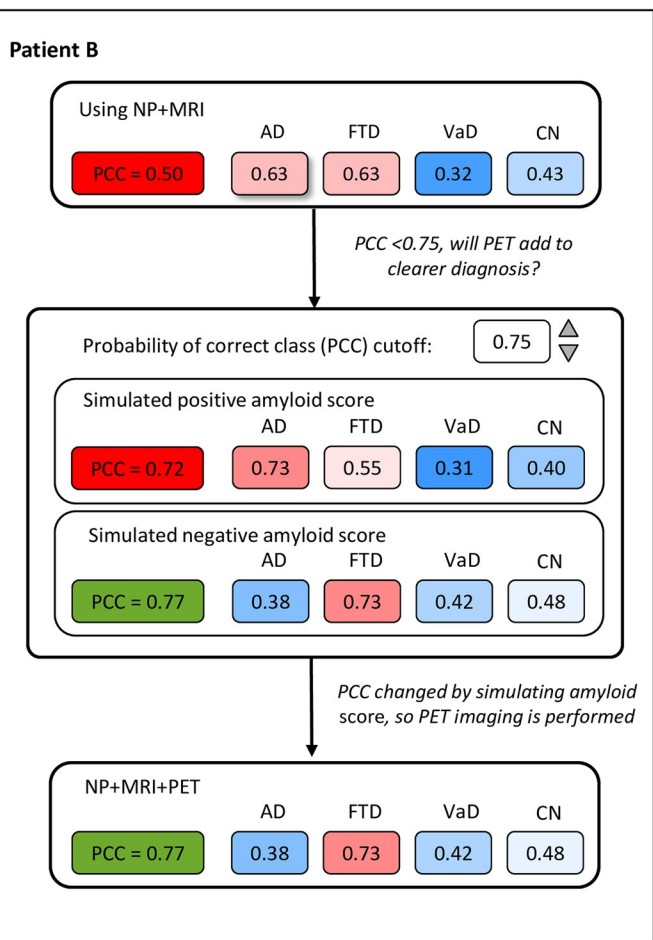

**Fig 3. Examples of visualization of the computerized decision approach for clinical use, applying hypothetical positive and negative amyloid-PET scan, based on visual reads.** NP: neuropsychology, MRI: magnetic resonance imaging, PCC: probability of correct class, AD: Alzheimer's disease, FTD: Frontotemporal dementia, VAD: Vascular dementia, CN: control.

PET is deemed appropriate in patients with possible AD 'for whom substantial uncertainty exists and for whom greater confidence would result from determining whether amyloid pathology is present or not', and in young-onset dementia to increase diagnostic confidence [13]. How to operationalize these criteria is not clear. Severable studies have shown that the current AUC advises amyloid-PET both too few and too many patients [4, 14, 16, 41]. Even in our study, 40% of the patients would require amyloid-PET, according to the AUC, without leading to higher proportion of patients with certain diagnosis.

Several prediction models predicting positive or negative amyloid-PET scans have been developed [42–44]. We add to this literature by developing a data-driven method with a different starting point. Namely, what happens to the diagnosis if an amyloid-PET is normal or abnormal? This approach follows the way clinicians think more naturally; 'would ordering an amyloid-PET scan help me in gaining a clearer and more certain diagnosis?'. We simulated a positive (AD-like) and negative (normal) amyloid-PET to estimate whether knowledge of amyloid status might impact (confidence in) diagnosis in an unselected memory clinic population, including controls, AD, FTD, VaD patients. Our computerized approach(scenario A) led to 152 (53%) correctly classified subjects while performing amyloid-PET in only 60 (21%)

subjects. Performing PET in all (scenario D) led to 142 (49%) correctly classified subjects, yet by performing amyloid-PET in 286 (100%) subjects. As can be seen in Fig 1, accuracy is slightly higher in D, since overall less patients received a diagnosis in this approach. One can imagine that in case of multiple pathologies or borderline amyloid-PET results, adding amyloid-PET only confuses and leads thus to a lower number of certainly diagnosed patients. These findings show that it is possible to think of scenario's where expensive diagnostic tests are used only when they are likely to increase diagnostic certainty, which is in line with appropriate use criteria stating that an additional test should only be performed when it will increase the confidence of the clinician in a certain diagnosis.

As the prevalence of dementia increases and new disease-modifying therapies (DMTs) entering, there is an increasing need for precise etiological diagnosis, while the diagnostic work-up needs to remain efficient [45]. To initiate treatment with DMTs, an accurate etiological diagnosis is crucial [46].In this future clinical practice, where DMTs are widely available [47], a data-driven approach will serve as a valuable tool for narrowing the target population for treatment. Our study presents a data-driven approach that is aimed at achieving diagnosis in the most efficient manner without compromising diagnostic performance. In addition such approach may streamline clinical decision-making pipelines with blood-based biomarkers in the future by limiting the number of patients that require confirmatory testing [48]. However, detecting underlying (AD) pathology marks only the beginning. Once the diagnose is made, the subsequent step will be to define eligibility. In a new EU-project, we will further develop this stepwise approach to identify potential eligible patients [49]. This approach will encompass other patients as well, such as those with mild cognitive impairment (MCI) and DLB, to address the relevant question whether they have underlying AD or not. Novel decision models have to be developed to aid in this classification, of which work is ongoing.

The classifier used in this study, is based on simple supervised machine learning and is thus able to deal with missing data. Providing visualization of the approach, with a PCC cut-off that clinicians can alter, as in Fig 3, helps clinicians to understand what the tool 'thinks', as opposed to a black box [9]. Visualization is also helpful in shared decision-making, guiding clinician and patient in discussing whether to perform amyloid-PET [50]. The visualization we have shown here can be further optimized in co-creation with end-users and usability testing in clinical practice. The cut-off of 0.75 for the PCC was selected for this manuscript to demonstrate how the proposed computerized decision support algorithm typically performs. If the clinician prefers higher classification accuracy with less patients diagnosed, a higher cut-off should be used, and vice versa. To date this method is not yet available for clinical use, but the previously developed data driven approach to select the optimal patient for CSF [18] is available via the cNeuro® tool.

Finally, the strengths of our study are the use of an unselected memory clinic cohort, consisting of controls and patients with AD, FTD and VaD, reflecting clinical practice [51].

There are also limitations to discuss. First, we were not able to include other neurodegenerative diseases, such as DLB, or those with MCI, since the used classifier to date does not include these patients. Tauopathies mimicking AD, such as argyrophillic grain disease (AGD) are not in our database, and could therefore not be included. To reflect even better ordinary clinical practice, development of the classifier is ongoing to include more diagnostic groups [52]. However, this study was set up to address the use of amyloid PET in differential diagnosis of a number of common differential diagnostic dilemma's, in particular AD versus FTD versus VaD. Second, we included only patients from a tertiary memory clinic, which may hamper generalizability, yet also reflects daily practice since amyloid-PET is mainly ordered in tertiary memory clinics. Third, we classified patients as having one diagnosis, while patients seldom have one type of neurodegeneration but often comorbid pathology. Yet, the DSI classifier

provides room for comorbid pathology by providing a DSI for each diagnosis, where co-existing pathologies would lead to multiple diagnoses with comparable DSI, while pure disease would results in one DSI standing out compared to the others. Also, in this cohort, comorbid pathology was present, given the often small differences in DSI between the first and second suggested diagnosis. In addition, amyloid-PET is not as specific as tau-PET, leading to more frequent discordance with the clinical diagnosis, which was also the case in our cohort. Fourth, while all patients received a standardized workup and were scanned with the same PET scanner, the MRI scanners differed. Yet we know from previous studies that our MRI quantification tool can deal with different scanners and field strengths [32]. Finally, we performed our analyses with visual readings being a dichotomize measure, which could be a disadvantage. However, visual readings are most often used in clinical practice and thus are easy for clinicians to understand. In addition, repeating our analyses with continuous values, namely Centiloids, a linear transformation off SUV, showed comparable results. This shows the face validity of our results.

## Conclusion

With the current difficulties in selecting those who might benefit from amyloid-PET and the future challenges with increasing need for biomarker confirmation, for example in the context of initiating disease modifying treatment, smart tools are needed to efficiently use resources and keep healthcare affordable. We developed a data-driven approach using patient's data and show that restricting the ordering of amyloid-PET to 21% of patients without compromising diagnostic performance. Future studies focusing on implementing tools like this into clinical practice, to efficiently guide stepwise diagnostic testing, are the next step [49].

## Supporting information

**S1 Fig. Flow chart for the four diagnostic approaches, using Centiloid values, summarizing the results in the last column.** AUC: appropriate use criteria, AUC+: patients fulfilling appropriate use criteria according to [13], operationalized as described in [14], PCC: probability of correct class, NP: neuropsychology, MRI: magnetic resonance imaging, Sim: simulate, FU: follow-up.
(PPTX)

## Acknowledgments

Research of the Alzheimer's Center Amsterdam is part of the neurodegeneration research program of Amsterdam Neuroscience. We thank Mahnaz Shekari for her significant contribution to the processing of the amyloid-PET images.

## Author Contributions

**Conceptualization:** Hanneke F. M. Rhodius-Meester, Ingrid S. van Maurik, Wiesje M. van der Flier.

**Data curation:** Frederik Barkhof, Elsmarieke van de Giessen.

**Formal analysis:** Hanneke F. M. Rhodius-Meester, Ingrid S. van Maurik, Lyduine E. Collij, Juha Koikkalainen, Antti Tolonen.

**Methodology:** Hanneke F. M. Rhodius-Meester, Ingrid S. van Maurik, Lyduine E. Collij, Juha Koikkalainen, Antti Tolonen, Johannes Berkhof, Frederik Barkhof, Jyrki Lötjönen.

**Project administration:** Hanneke F. M. Rhodius-Meester.

**Resources:** Aniek M. van Gils.

**Software:** Lyduine E. Collij, Juha Koikkalainen, Elsmarieke van de Giessen, Jyrki Lötjönen.

**Supervision:** Yolande A. L. Pijnenburg, Jyrki Lötjönen, Wiesje M. van der Flier.

**Validation:** Juha Koikkalainen.

**Visualization:** Hanneke F. M. Rhodius-Meester, Ingrid S. van Maurik.

**Writing – original draft:** Hanneke F. M. Rhodius-Meester.

**Writing – review & editing:** Ingrid S. van Maurik, Lyduine E. Collij, Aniek M. van Gils, Juha Koikkalainen, Antti Tolonen, Yolande A. L. Pijnenburg, Johannes Berkhof, Frederik Barkhof, Elsmarieke van de Giessen, Jyrki Lötjönen, Wiesje M. van der Flier.

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
