## [Decision Letter · Decision Letter 0]

14 Dec 2023

PONE-D-23-25666Computerized decision support is an effective approach to select memory clinic patients for amyloid-PETPLOS ONE

Dear Dr. Rhodius-Meester,

Thank you for submitting your manuscript to PLOS ONE. After careful consideration, we feel that it has merit but does not fully meet PLOS ONE’s publication criteria as it currently stands. Therefore, we invite you to submit a revised version of the manuscript that addresses the points raised during the review process.

The reviewers and this editor raise criticisms and concerns about the contents of the manuscript, as described below. Especially one reviewer critically questions what the merit or advantage of this study is in daily clinical practice. The authors should include clear responses to these criticisms.

We look forward to receiving your revised manuscript.

Kind regards,

Wataru Araki

Academic Editor

PLOS ONE

Journal Requirements:

"Research of the Alzheimer’s Center Amsterdam is part of the neurodegeneration research program of Amsterdam Neuroscience. The Vrije Universiteit Medical Center Alzheimer Center is supported by the Stichting Alzheimer Nederland and Stichting Vrije Universiteit Medical Center Fonds. The clinical database structure was developed with funding from Stichting Dioraphte. For development of the PredictAD tool, VTT Technical Research Centre of Finland has received funding from European Union’s Seventh Framework Programme for research, technological development and demonstration under grant agreements601055 (VPH-DARE@IT) ), 224328 (PredictAD), and 611005 (PredictND). The collaboration project DAILY (project number LSHM19123-HSGF) is co-funded by the PPP Allowance made available by Health-Holland, Top Sector Life Sciences & Health, to stimulate public-private partnerships. The ABIDE clinical utility study is funded by the PPP Allowance made available by health-Holland, Top Sector Life Sciences & Health, to stimulate public-private partnerships and co-funded by Life Molecular Imaging GmbH (grant no.: LSHM18075). HR is recipient of the Memorabel Dementia Fellowship 2021 (ZonMw projectnumber 10510022110004) and Alzheimer Nederland InterACT grant (projectnumber WE.08-2022-06). LC is supported by the Innovative Medicines Initiative 2 Joint Undertaking under grant agreement No 115952. This Joint Undertaking receives the support from the European Union’s Horizon 2020 research and innovation program and EFPIA. FB is supported by the NIHR biomedical research centre at UCLH. The chair of WF is supported by the Pasman Stichting. WF is recipient of ABOARD, which is a public–private partnership receiving funding from ZonMW (number 73305095007) and Health-Holland, Top Sector Life Sciences and Health (public–private partnership allowance; number LSHM20106), WF and HR are recipient of the Horizon 2022 project PROMINENT (project number 101112145). We thank Mahnaz Shekari for her significant contribution to the processing of the amyloid-PET images."

"We have read the journal's policy and the authors of this manuscript have the following competing interests: Hanneke FM Rhodius- Meester performs contract research for Combinostics, all funding is paid to her institution. Ingrid van Maurik received a consultancy fee (paid to the university) from Roche. Lyduine E Collij has received consultancy fees from GE Healthcare, all funding is paid to her institution. Aniek M van Gils reports no disclosures. Juha Koikkalainen and Jyrki Lötjönen report that VTT Technical Research Centre of Finland owns the following IPR related to the paper: 1. J. Koikkalainen and J. Lotjonen. A method for inferring the state of a system, US7,840,510 B2, PCT/FI2007/050277. 2. J. Lotjonen, J. Koikkalainen and J. Mattila.State Inference in a heterogeneous system, PCT/FI2010/050545. FI20125177. Koikkalainen and Lötjönen are shareholders in Combinostics Oy. Antti Tolonen reports no disclosures. Yolande AL Pijnenburg has received funding from Dioraphte Foundation, Zabawas Foundation, JPND, ZonMW, NWO, Team Alzheimer and the Dutch Brain Foundation. Johannes Berkhof reports no disclosures. Frederik Barkhof is member of the Steering committee or Data Safety Monitoring Board member for Biogen, Merck, ATRI/ACTC and Prothena. FB is consultant for Roche, Celltrion, Rewind Therapeutics, Merck, IXICO, Jansen, Combinostics. FB has research agreements with Merck, Biogen, GE Healthcare, Roche. Co-founder and shareholder of Queen Square Analytics LTD. Elsmarieke van de Giessen has received research support from NWO, ZonMw, Hersenstichting and KWF. EvdG has performed contract research for Heuron Inc., Roche and 1st Biotherapeutics. EvdG has a consultancy agreement with IXICO for the reading of PET scans. Wiesje M van der Flier performs contract research for Biogen. Research programs of WF have been funded by ZonMW, NWO, EU-FP7, EU-JPND, Alzheimer Nederland, CardioVascular Onderzoek Nederland, Health~Holland, Topsector Life Sciences & Health, stichting Dioraphte, Gieskes-Strijbis fonds, stichting Equilibrio, Pasman stichting, stichting Alzheimer & NeuroPsychiatry Foundation, Philips, Biogen MA Inc, Novartis-NL, Life-MI, AVID, Roche BV, Fujifilm, Combinostics. WF has performed contract research for Biogen MA Inc, and Boehringer Ingelheim. WF has been an invited speaker at Boehringer Ingelheim, Biogen MA Inc, Danone, Eisai, WebMD Neurology (Medscape), Springer Healthcare. WF is consultant to Oxford Health Policy Forum CIC, Roche, and Biogen MA Inc. WF participated in advisory boards of Biogen MA Inc and Roche. All funding is paid to her institution. WF was associate editor of Alzheimer, Research & Therapy in 2020/2021. WF is associate editor at Brain"

6. We note that you have indicated that data from this study are available upon request. PLOS only allows data to be available upon request if there are legal or ethical restrictions on sharing data publicly. For more information on unacceptable data access restrictions, please see http://journals.plos.org/plosone/s/data-availability#loc-unacceptable-data-access-restrictions.  

7. We note that you have stated that you will provide repository information for your data at acceptance. Should your manuscript be accepted for publication, we will hold it until you provide the relevant accession numbers or DOIs necessary to access your data. If you wish to make changes to your Data Availability statement, please describe these changes in your cover letter and we will update your Data Availability statement to reflect the information you provide.

Additional Editor Comments :

In addition to the reviewers’ comments, this editor considers that several parts in the manuscript need to be clarified or explained in more details, as follows.

1 p8 line 230-233 They cite ref13 but citing this ref. is insufficient. Please explain about AUC in more details.

2 Subjects in page 4 and Discussion

They included patients with AD, FTD, VaD, and SCD. Please explain why they did not include those with other dementia disorders such as DLB and AGD. In ordinary clinical practice, these non-AD dementia disorders are relatively common and should not be neglected. Clearly, this is a limitation of this study and should also be discussed.

3 Page 7, Ref 37 A correct ref should be described.

4 Fig.2 This figure should be corrected as it does not cover the whole graphs. Fig 3 Is it possible to change the background color from blue to white?

5 Discussion page 14 line 403-405 “Our computerized approach even outranked performing 404 amyloid-PET in all patients” Please explain about the reasons for this result more clearly.

6. Discussion page 15 line 418-431

This editor judges that this part is not a balanced view and should be shortened or more carefully revised. It can only be said that a data-driven approach will be a supportive tool in the future clinical practice in which disease-modifying treatments become widely available. In addition, in the clinical practice, differential diagnosis of MCI patients (MCI due to AD or MCI not due to AD) is also important but this issue is not discussed in the manuscript. Why not include some discussion about this point?

7 Abstract line 48 “AUC” Please include statement about what AUC means.

8 Abstract the last sentence “ supports clinicians in making a balanced decision in ordering additional amyloid PET during the dementia workup” This does not reflect the real situation, and should be corrected.

Reviewers' comments:

Reviewer's Responses to Questions

**Comments to the Author**

1. Is the manuscript technically sound, and do the data support the conclusions?

Reviewer #1: Partly

Reviewer #2: Yes

2. Has the statistical analysis been performed appropriately and rigorously? 

Reviewer #1: N/A

Reviewer #2: Yes

3. Have the authors made all data underlying the findings in their manuscript fully available?

Reviewer #1: No

Reviewer #2: Yes

4. Is the manuscript presented in an intelligible fashion and written in standard English?

Reviewer #1: Yes

Reviewer #2: Yes

5. Review Comments to the Author

Reviewer #1: With the insurance coverage of Lecanemab and other therapeutic agents against amyloid-β, there is growing interest in testing for amyloid-β detection. As the authors state in their introduction, PET scans are too expensive to be used for screening, raising the question of which patients should be aggressively screened for PET.

The purpose of this study is to "develop a computerized decision support approach to select patients for amyloid PET," but the study targets AD, FTLD, and VaD, and is more focused on disease type diagnosis rather than DMT target selection. Therefore, the study is disappointing to those who expect it to be a supportive approach to narrow down the target population for treatment of amyloid-β.

At the method of study, the following details should be considered.

1. the final classification result (likelihood of diagnosis) seems to be obtained by combining binary classification by DSI, but the rationale for "we considered that patients had a sufficiently reliable diagnosis if the PCC was 0.75 or higher" is not provided. Presumably, there is a valid basis, but it is not clear how many false positives or false negatives this may include.

2. it says that DSI "can handle missing data," but it is not clear how much missingness is actually allowed or how much it affects accuracy.

3. Figure 3 is helpful in understanding this study. For example, in Patient B, FTD is suspected and an amyloid PET scan is recommended as a result. However, without presentation of the present illness, symptoms, results of psychological testing, and MRI findings, the usefulness of DSI cannot be realized for readers. Wouldn't ordinary clinical diagnostic methods have been sufficient to suspect FTD?

4. it is stated that "the strength of this study is the use of a control group and an unselected memory clinic cohort consisting of AD, FTD, and VaD patients, which reflects clinical practice," but it is not stated how PD, epilepsy, and other illness were excluded to get there.

First of all, I would like to express my utmost respect for the results of this study. In general, the study would have been more interesting if it had been conducted with DMT in mind. In understanding the utility of DSI, which requires the input of a variety of clinical data, what is its advantage over the diagnoses we clinicians make in our daily clinical practice? If we could understand how DSI differs from and is superior to the diagnosis that we clinicians usually make in daily clinical practice, it would be more convincing.

Reviewer #2: The article of Hanneke et al. affirms that a computerized decision support approach to select patients from the Memory Clinic for amyloid-PET increase the diagnostic certainty, being useful for the doctor requesting. The approach applies The Disease State Index (DSI) classifier, already tested with proven high diagnostic accuracy [references 35 and 36 of the authors] . The DSI suggests a diagnosis with a probability of correct class (PCC), based on demographics, neuropsychology, and MRI data. Thereafter, the hypothetical positive and negative amyloid-PET PCC’s values are calculated to evaluate the influence in the diagnostic certainty. The authors point out that this starting point makes the difference to other models that predict positive or negative amyloid-PET [references 40-42 of the authors]. Afterwards, an amyloid PET scan is order only if an increase to >0.75 of the PCC. Finally, the real amyloid-PET values are added to the model and PCC is evaluated.

This approach is compared with three control scenarios: without amyloid- PET using demographics, APOE, neuropsychology, and MRI; applying amyloid-PET based on The Appropriate Use Criteria (AUC) [reference 12 of the authors]; and performing amyloid-PET in all patients. Using visual PET reads and Centiloid values submit similar results to support the affirmation as a meaningful conclusion.

This proposed tool is quite relevant for the clinics, as the authors justify well in the introduction and discussion, considering nowadays there is conclusive evidence on the clinical usefulness of amyloid-PET, in the in vivo diagnosis of Alzheimer's disease (AD), as shown the results of two large international series [the American project IDEAS: reference 4 of the authors; and the European AMYPAD: reference 7 of the authors and Altomare et al. 2023]. Furthermore, even if the amyloid-PET is prescribed regarding the AUC there are other situations in which the added certainty of amyloid-PET could be helpful [reference 13 of the authors and Altomare et al. 2018]. Moreover, an increasing demand [Verger et al. 2023] is predictably because of its contribution to AD diagnosis, AD treatment (the search and introduction of possible modifying therapies [Garnier-Crussard A et al. 2023] and is the gold standard for investigating disease mechanisms [Bao et al. 2021]) and to screening improvement for clinical trials [Rabinovici et al. 2019].In words of the authors: amyloid-PET is costly and limitedly available, so intelligent and efficient use of our resources is already needed.

The statistics, with a large sample size (N=286) including controls (N=135), and other analyses (v.g. amyloid-PET procedure, Centiloid), are well conducted with a high technical standard and are described to enable reproduce. Results are exposed with sufficient detail (including mean and standard deviation) represented with clear figures and tables. The manuscript is properly presented and is written in standard English. The research meets all applicable experimentation ethics standards and integrity (blinding etc.). The article adheres to appropriate reporting guidelines and community standards for data availability.

In this work, a real-life cohort is studied (recruiting in a tertiary memory clinic) in contradistinction to the computerized decision support approach applied by the author of correspondence et al. in the cohort of Alzheimer’s Disease Neuroimaging Initiative (ADNI2) that can be consulted at Alzheimer’s Dement. 2020;16(Suppl. 5):e042687).Therefore, it cannot be considered as a replication study. The data were collected during routine care and the amyloid-PET is interpreted visually by an imaging specialist from the local hospital setting, as often do in practice and in the IDEAS project [reference 4 of the authors], and no by centralizing reading. Agreeing to the authors, this naturalistic setting features, far from being a limitation, represent an advantage due to its immediate applicability in clinical practice in the dementia workup. The strengths and limitations are well exposed.

Some aspects that may weaken the quality of the manuscript and that the authors could clarify so readers better understand could be: DSI is not included in the abstract, the data of the patients correctly classified presented in the flow chart (Figure 1) seems no be present or explained in the text. Also review: line 283 doubt about the 180 patients (66%)¿is it maybe 188 (66%)?, line 293-297 review typographical errors and also doubt about 1654 (54%).%)¿is it maybe 154(54%)?, line 638: 37. !!! INVALID CITATION !!! [20, 36]. Please, take as a kind suggestion the possibility of including the refrence of AMYPAD 2023 in the introduction and update the reference 44. Maybe will be interesting a brief mention about the availability of the computerized decision support approach created by the authors for use in other centers.

References:

Altomare D, Barkhof F, Caprioglio C, Collij LE, Scheltens P, Lopes Alves I, et al. Clinical effect of early vs late amyloid positron emission tomography in Memory Clinic patients: The AMYPAD-DPMS randomized clinical trial. JAMA Neurology [Internet]. 2023 May 8 Available from: https://jamanetwork.com/journals/jamaneurology/fullarticle/2804755

Altomare D, Ferrari C, Festari C, Guerra UP, Muscio C, Padovani A, et al. Quantitative appraisal of the Amyloid Imaging Taskforce Appropriate Use Criteria for amyloid‐PET. Alzheimers Dement. 2018;14(8):1088-98.

Verger A, Yakushev I, Albert NL, Van Berckel B, Brendel M, Cecchin D, et al. FDA approval of lecanemab: the real start of widespread amyloid PET use? — the EANM Neuroimaging Committee perspective. Eur J Nucl Med Mol Imaging. 2023;50(6):1553-5.

Garnier-Crussard A, Flaus A. Positive opinion of the French National Authority for Health on the reimbursement of amyloid tracer (Flutemetamol). Eur J Nucl Med Mol Imaging. 2023;50(2):253-4.

Bao W, Xie F, Zuo C, Guan Y, Huang YH. PET Neuroimaging of Alzheimer’s Disease: radiotracers and their utility in clinical research. Front Aging Neurosci. 2021; 13:624330.

Rabinovici GD, Gatsonis C, Apgar C, Chaudhary K, Gareen I, Hanna L, et al. Association of amyloid positron emission tomography with subsequent change in clinical management among Medicare beneficiaries with mild cognitive impairment or dementia. JAMA. 2019;321(13):1286.

6. PLOS authors have the option to publish the peer review history of their article (what does this mean?). If published, this will include your full peer review and any attached files.

Reviewer #1: No

Reviewer #2: **Yes: **RAQUEL SÁNCHEZ VAÑÓ

---

## [Author Response · Author response to Decision Letter 0]

26 Jan 2024

Amsterdam, January 26th 2024

Regarding PONE-D-23-25666

Dear dr Wataru Araki, Academic Editor, and esteemed Reviewers of PlosOne. 

Please find uploaded our revised manuscript for publication in PLOS ONE ‘Computerized decision support is an effective approach to select memory clinic patients for amyloid-PET’. 

We thank the editor and reviewers for their careful reading, thoughtful comments, and recommendation for revision. Please find below our responses to the comments in a point-by-point fashion. We have highlighted changes (via track&change) in response to the reviewers’ comments in the manuscript. 

We hope the rebuttal adequately addresses the points raised during the review process, 

Kind regards, also on behalf of the co-authors, 

Hanneke Rhodius- Meester

Journal Requirements:

- Reply: To our knowledge our manuscript meets the style requirements, but we are happy to make changes if needed. 

- Reply: We are not sure to which code you are referring in our manuscript. Syntaxes used have been created in MatLab and have been described in previous publication (Cluitmans et al 2013 A MATLAB toolbox for classification and visualization of heterogenous multi-scale human data using the Disease State Fingerprint method - PubMed (nih.gov)), as stated in the methods section (L265-266). 

"Research of the Alzheimer’s Center Amsterdam is part of the neurodegeneration research program of Amsterdam Neuroscience. The Vrije Universiteit Medical Center Alzheimer Center is supported by the Stichting Alzheimer Nederland and Stichting Vrije Universiteit Medical Center Fonds. The clinical database structure was developed with funding from Stichting Dioraphte. For development of the PredictAD tool, VTT Technical Research Centre of Finland has received funding from European Union’s Seventh Framework Programme for research, technological development and demonstration under grant agreements601055 (VPH-DARE@IT) ), 224328 (PredictAD), and 611005 (PredictND). The collaboration project DAILY (project number LSHM19123-HSGF) is co-funded by the PPP Allowance made available by Health-Holland, Top Sector Life Sciences & Health, to stimulate public-private partnerships. The ABIDE clinical utility study is funded by the PPP Allowance made available by health-Holland, Top Sector Life Sciences & Health, to stimulate public-private partnerships and co-funded by Life Molecular Imaging GmbH (grant no.: LSHM18075). HR is recipient of the Memorabel Dementia Fellowship 2021 (ZonMw projectnumber 10510022110004) and Alzheimer Nederland InterACT grant (projectnumber WE.08-2022-06). LC is supported by the Innovative Medicines Initiative 2 Joint Undertaking under grant agreement No 115952. This Joint Undertaking receives the support from the European Union’s Horizon 2020 research and innovation program and EFPIA. FB is supported by the NIHR biomedical research centre at UCLH. The chair of WF is supported by the Pasman Stichting. WF is recipient of ABOARD, which is a public–private partnership receiving funding from ZonMW (number 73305095007) and Health-Holland, Top Sector Life Sciences and Health (public–private partnership allowance; number LSHM20106), WF and HR are recipient of the Horizon 2022 project PROMINENT (project number 101112145). We thank Mahnaz Shekari for her significant contribution to the processing of the amyloid-PET images."

Please include your amended statements within your cover letter; we will change the online submission form on your behalf:

- Reply: Apologies for misunderstanding this section, we have added amended statements to our cover letter: 

Acknowledgment Section: "Research of the Alzheimer’s Center Amsterdam is part of the neurodegeneration research program of Amsterdam Neuroscience. The Vrije Universiteit Medical Center Alzheimer Center is supported by the Stichting Alzheimer Nederland and Stichting Vrije Universiteit Medical Center Fonds. The clinical database structure was developed with funding from Stichting Dioraphte. We thank Mahnaz Shekari for her significant contribution to the processing of the amyloid-PET images.

Funding Statement: “For development of the PredictAD tool, VTT Technical Research Centre of Finland has received funding from European Union’s Seventh Framework Programme for research, technological development and demonstration under grant agreements601055 (VPH-DARE@IT) ), 224328 (PredictAD), and 611005 (PredictND). The collaboration project DAILY (project number LSHM19123-HSGF) is co-funded by the PPP Allowance made available by Health-Holland, Top Sector Life Sciences & Health, to stimulate public-private partnerships. The ABIDE clinical utility study is funded by the PPP Allowance made available by health-Holland, Top Sector Life Sciences & Health, to stimulate public-private partnerships and co-funded by Life Molecular Imaging GmbH (grant no.: LSHM18075). HR is recipient of the Memorabel Dementia Fellowship 2021 (ZonMw projectnumber 10510022110004) and Alzheimer Nederland InterACT grant (projectnumber WE.08-2022-06). LC is supported by the Innovative Medicines Initiative 2 Joint Undertaking under grant agreement No 115952. This Joint Undertaking receives the support from the European Union’s Horizon 2020 research and innovation program and EFPIA. FB is supported by the NIHR biomedical research centre at UCLH. The chair of WF is supported by the Pasman Stichting. WF is recipient of ABOARD, which is a public–private partnership receiving funding from ZonMW (number 73305095007) and Health-Holland, Top Sector Life Sciences and Health (public–private partnership allowance; number LSHM20106), WF and HR are recipient of the Horizon 2022 project PROMINENT (project number 101112145).”

Thank you for changing the online submission form. 

"We have read the journal's policy and the authors of this manuscript have the following competing interests: Hanneke FM Rhodius- Meester performs contract research for Combinostics, all funding is paid to her institution. Ingrid van Maurik received a consultancy fee (paid to the university) from Roche. Lyduine E Collij has received consultancy fees from GE Healthcare, all funding is paid to her institution. Aniek M van Gils reports no disclosures. Juha Koikkalainen and Jyrki Lötjönen report that Combinostics Oy owns the following IPR related to the paper: 1. J. Koikkalainen and J. Lotjonen. A method for inferring the state of a system, US7,840,510 B2, PCT/FI2007/050277. 2. J. Lotjonen, J. Koikkalainen and J. Mattila.State Inference in a heterogeneous system, PCT/FI2010/050545. FI20125177. Koikkalainen and Lötjönen are shareholders in Combinostics Oy. Antti Tolonen reports no disclosures. Yolande AL Pijnenburg has received funding from Dioraphte Foundation, Zabawas Foundation, JPND, ZonMW, NWO, Team Alzheimer and the Dutch Brain Foundation. Johannes Berkhof reports no disclosures. Frederik Barkhof is member of the Steering committee or Data Safety Monitoring Board member for Biogen, Merck, ATRI/ACTC and Prothena. FB is consultant for Roche, Celltrion, Rewind Therapeutics, Merck, IXICO, Jansen, Combinostics. FB has research agreements with Merck, Biogen, GE Healthcare, Roche. Co-founder and shareholder of Queen Square Analytics LTD. Elsmarieke van de Giessen has received research support from NWO, ZonMw, Hersenstichting and KWF. EvdG has performed contract research for Heuron Inc., Roche and 1st Biotherapeutics. EvdG has a consultancy agreement with IXICO for the reading of PET scans. Wiesje M van der Flier performs contract research for Biogen. Research programs of WF have been funded by ZonMW, NWO, EU-FP7, EU-JPND, Alzheimer Nederland, CardioVascular Onderzoek Nederland, Health~Holland, Topsector Life Sciences & Health, stichting Dioraphte, Gieskes-Strijbis fonds, stichting Equilibrio, Pasman stichting, stichting Alzheimer & NeuroPsychiatry Foundation, Philips, Biogen MA Inc, Novartis-NL, Life-MI, AVID, Roche BV, Fujifilm, Combinostics. WF has performed contract research for Biogen MA Inc, and Boehringer Ingelheim. WF has been an invited speaker at Boehringer Ingelheim, Biogen MA Inc, Danone, Eisai, WebMD Neurology (Medscape), Springer Healthcare. WF is consultant to Oxford Health Policy Forum CIC, Roche, and Biogen MA Inc. WF participated in advisory boards of Biogen MA Inc and Roche. All funding is paid to her institution. WF was associate editor of Alzheimer, Research & Therapy in 2020/2021. WF is associate editor at Brain"

- Reply: Thank you for this addition. We added an update Competing Interests statement to our cover letter, stating "This does not alter our adherence to PLOS ONE policies on sharing data and materials.”.

- Reply: We added the requested information on the consent procedure to the paragraph ‘Subjects’ in the Material and Methods section (L121-124).

6. We note that you have indicated that data from this study are available upon request. PLOS only allows data to be available upon request if there are legal or ethical restrictions on sharing data publicly. For more information on unacceptable data access restrictions, please see http://journals.plos.org/plosone/s/data-availability#loc-unacceptable-data-access-restrictions. 

- Reply: Thank you for addressing this important point. We are afraid there are indeed restrictions to data-sharing, since the data contains potentially sensitive information (namely medical data). In the patient information folder (PIF), that subjects read before giving written informed consent, it is therefore stated specifically that their data is only shared with third parties when that data is used to gain more knowledge on Alzheimer’s disease and dementia. We can only guarantee this, via a data-request form and checking if the intended use of data aligns with the consent given by subjects. The data underlying the results presented in this study are therefore available upon request from the Data access team of the Alzheimer Center Amsterdam, via wm.vdflier@amsterdamumc.nl. We added this statement to our cover letter. 

7. We note that you have stated that you will provide repository information for your data at acceptance. Should your manuscript be accepted for publication, we will hold it until you provide the relevant accession numbers or DOIs necessary to access your data. If you wish to make changes to your Data Availability statement, please describe these changes in your cover letter and we will update your Data Availability statement to reflect the information you provide.

- Reply: Apologies to this misunderstanding. The data used in this study is part of the Amsterdam Dementia Cohort, of the Alzheimer Center Amsterdam, and can be retrieved from there via a data request. More information on the ADC can be found at JPND, Amsterdam dementia cohort - JPND Neurodegenerative Disease Research (neurodegenerationresearch.eu), unfortunately without a DOI. We have updated our Data Availability statement in the cover later. Thank you for updating our statement. 

Additional Editor Comments :

In addition to the reviewers’ comments, this editor considers that several parts in the manuscript need to be clarified or explained in more details, as follows.

1 p8 line 230-233 They cite ref13 but citing this ref. is insufficient. Please explain about AUC in more details.

- Reply: Thank you for the opportunity to explain how we defined patients to be AUC positive or negative. De Wilde et al (ref 13) previously classified each included patient. In short, clinical syndrome (dementia, MCI, or SCD), suspected etiology (AD, vascular pathology, frontotemporal dementia, Lewy body dementia, other neurodegenerative disease, or non-neurodegenerative disease), and level of diagnostic confidence in suspected etiology (visual analog scale, 0–100%) were determined during pre-PET multidisciplinary meetings. After that patients were classified as AUC+ when they either i) had AD as diagnostic possibility (≥15%) but with a confidence <85% in AD as diagnosis, or ii) had a young-onset dementia (<65 years old. All other patients were classified as AUC-.This method was based on Altomare et al in 2018 10.1016/j.jalz.2018.02.022. We added details on this in the Material and Methods section (L243-246) and the Abstract (L34-36).

2 Subjects in page 4 and Discussion

They included patients with AD, FTD, VaD, and SCD. Please explain why they did not include those with other dementia disorders such as DLB and AGD. In ordinary clinical practice, these non-AD dementia disorders are relatively common and should not be neglected. Clearly, this is a limitation of this study and should also be discussed.

- Reply: Unfortunately we were not able to include other neurodegenerative diseases, such as dementia with Lewy bodies, or MCI (

---

## [Decision Letter · Decision Letter 1]

5 Mar 2024

PONE-D-23-25666R1Computerized decision support is an effective approach to select memory clinic patients for amyloid-PETPLOS ONE

Dear Dr. Rhodius-Meester,

Thank you for submitting your manuscript to PLOS ONE. After careful consideration, we feel that it has merit but does not fully meet PLOS ONE’s publication criteria as it currently stands. Therefore, we invite you to submit a revised version of the manuscript that addresses the points raised during the review process.

This editor judges that some parts of the revised manuscript still need to be clarified, as noted in the comments of the editor. 

We look forward to receiving your revised manuscript.

Kind regards,

Wataru Araki

Academic Editor

PLOS ONE

Journal Requirements:

Additional Editor Comments:

In the revised manuscript, the authors have addressed almost all the criticisms of the reviewers. However, this editor thinks that some parts are not well explained or clarified and still need to be amended as indicated below. Especially, the authors should be careful to make the manuscript more comprehensible in general.

1) Response to Editor comment 5

L405-406, L428-430? L523?

These revisions are not enough to clarify the question raised by the editor.

2) Response to Reviewer #1 comment 4 and Editor comment 2

L119 is not enough. Please explain about the reason for this point.

3) Discussion L427-429

As such, -----testing [45]. This sentence seems to be an overstatement. Please change the expression.

Discussion L431-438 It is better to move this paragraph after the first paragraph in Discussion.

Discussion L455-463 This paragraph is too superficial and may be omitted.

Discussion L478-500 This paragraph should be revised so that readers can understand what the limitations of this work are. Please list the limitations in order (First, second, third ----)

Line 482 a MCI should be MCI

Line 483-485 Recruiting in ----- value. This sentence is not understandable and should be rephrased.

Conclusion Line 510-511 “to funnel patient through the diagnostic pathway,” This expression is not grammatically correct and should be corrected.

Reviewers' comments:

Reviewer's Responses to Questions

**Comments to the Author**

1. If the authors have adequately addressed your comments raised in a previous round of review and you feel that this manuscript is now acceptable for publication, you may indicate that here to bypass the “Comments to the Author” section, enter your conflict of interest statement in the “Confidential to Editor” section, and submit your "Accept" recommendation.

Reviewer #2: All comments have been addressed

2. Is the manuscript technically sound, and do the data support the conclusions?

Reviewer #2: Yes

3. Has the statistical analysis been performed appropriately and rigorously? 

Reviewer #2: Yes

4. Have the authors made all data underlying the findings in their manuscript fully available?

Reviewer #2: Yes

5. Is the manuscript presented in an intelligible fashion and written in standard English?

Reviewer #2: Yes

6. Review Comments to the Author

Reviewer #2: (No Response)

7. PLOS authors have the option to publish the peer review history of their article (what does this mean?). If published, this will include your full peer review and any attached files.

Reviewer #2: **Yes: **RAQUEL SÁNCHEZ VAÑÓ

---

## [Author Response · Author response to Decision Letter 1]

15 Apr 2024

Additional Editor Comments:

In the revised manuscript, the authors have addressed almost all the criticisms of the reviewers. However, this editor thinks that some parts are not well explained or clarified and still need to be amended as indicated below. Especially, the authors should be careful to make the manuscript more comprehensible in general.

- Reply: We thank the editor for these suggestion and have changed the manuscript based on the comments below, and in general. 

1) Response to Editor comment 5. L405-406, L428-430? L523? These revisions are not enough to clarify the question raised by the editor. 

- Reply: The editor refers to the sentence in the original manuscript “Our computerized approach even outranked performing 404 amyloid-PET in all patients” where he during the first revision asked to explain the reasons for this result more clearly. In the revised version we rephrased this to “ Our approach was thus more efficient than the other scenario’s, where we would have performed PET in all patients, in none, or according to the appropriate use criteria (AUC).” in L405-407. 

We are not sure what the editor seeks to clarify. The stepwise approach (scenario A) leads to 152 (53%) correctly classified subjects performing only PET in 60 (21%) subjects. Performing PET in all (scenario D) leads to 142 (49%) correctly classified subjects, yet with PET in 286 (100%) subjects. As can be seen in Fig 1, the overall accuracy is slightly higher in D, since overall less patients received a diagnosis in the approach performing PET in all. This is why we rephrased the original sentence, replacing ‘outperformed’ to ‘more efficient’. We have now rephrased this in the abstract, throughout the entire discussion and added more explanation. 

Perhaps the editor questions why performing all tests in all is not superior to a stepwise approach? We feel our findings here are in line with several appropriate use criteria stating that an additional test should only be performed when it will increase the confidence of the clinician in a certain diagnosis. This was also added to the discussion. We hope this answer clarify the editor’s question. (L53-54, L59-60, L406, L425, L434-447, L464-465, L537, and we deleted sentence L538-539)

2) Response to Reviewer #1 comment 4 and Editor comment 2. L119 is not enough. Please explain about the reason for this point.

- Reply: The editor refers to his comment on “why they did not include those with other dementia disorders such as DLB and AGD. In ordinary clinical practice, these non-AD dementia disorders are relatively common and should not be neglected”, and the reviewer comment on “how PD, epilepsy, and other illness were excluded to get there.”

We totally agree that adding DLB would be useful. As explained in the methods, we were unfortunately not able to include other neurodegenerative diseases (such as DLB and AGD) or MCI, since the used classifier to date does not include these categories and therefore cannot classify them. And, though DLB is a common diagnosis in a memory clinic setting, the other diagnoses mentioned here are in fact less relevant. AGD, PD, epilepsy can sometimes occur in a memory clinic, yet are not common diagnoses (we unfortunately don’t even have AGD patients in our database) and do not pose a clinical dilemma very often. This study was set up to address the use of amyloid PET in differential diagnosis of a number of common differential diagnostic dilemma’s, in particular AD vs FTD vs VaD. We added more explanation about this to the methods section and to limitations in the discussion. We hope this explains our reason sufficiently. (L102, L118-120, L502-509)

3) Discussion L427-429. As such, -----testing [45]. This sentence seems to be an overstatement. Please change the expression.

-Reply: We apologize for this statement, which we added as a response to reviewer 1. We have now rephrased the sentence to ’ In addition, such approach may streamline clinical decision-making pipelines with blood-based biomarkers in the future by limiting the number of patients that require confirmatory testing [49]’. (L465-468).

4) Discussion L431-438 It is better to move this paragraph after the first paragraph in Discussion.

- Reply: We thank you for this suggestion. We have now moved this paragraph after the first paragraph, and deleted 3 sentences there to prevent doubling text. (L412-425)

5) Discussion L455-463 This paragraph is too superficial and may be omitted.

- Reply: We have now omitted this paragraph and changed the first sentence after this paragraph to allow for better flow. (L477-485, L487)

6) Discussion L478-500 This paragraph should be revised so that readers can understand what the limitations of this work are. Please list the limitations in order (First, second, third ----)

- Reply: We thank the editor for this suggestion, and have rewritten the limitations sections. (L500-523)

7) Line 482 a MCI should be MCI 

-- Reply: Apologies, we changed this into ‘MCI’. (L504)

8) Line 483-485 Recruiting in ----- value. This sentence is not understandable and should be rephrased.

- Reply: We rephrased the sentence to ‘Second, we included only patients from a tertiary memory clinic , which may hamper generalizability, yet also reflects daily practice since amyloid-PET is mainly ordered in tertiary memory clinics’. (L509-512)

9)Conclusion Line 510-511 “to funnel patient through the diagnostic pathway,” This expression is not grammatically correct and should be corrected.

- Reply: We rephrased this final sentence to ‘ to efficiently guide stepwise diagnostic testing’. (L540-541)

Reviewers' comments:

1. If the authors have adequately addressed your comments raised in a previous round of review and you feel that this manuscript is now acceptable for publication, you may indicate that here to bypass the “Comments to the Author” section, enter your conflict of interest statement in the “Confidential to Editor” section, and submit your "Accept" recommendation.

Reviewer #2: All comments have been addressed

2. Is the manuscript technically sound, and do the data support the conclusions?

Reviewer #2: Yes

3. Has the statistical analysis been performed appropriately and rigorously? 

Reviewer #2: Yes

4. Have the authors made all data underlying the findings in their manuscript fully available?

Reviewer #2: Yes

5. Is the manuscript presented in an intelligible fashion and written in standard English?

Reviewer #2: Yes

6. Review Comments to the Author

Reviewer #2: (No Response)

7. PLOS authors have the option to publish the peer review history of their article (what does this mean?). If published, this will include your full peer review and any attached files.

Reviewer #2: Yes: RAQUEL SÁNCHEZ VAÑÓ

Reply: we thank the reviewer for their efforts and are glad all comments have been addressed.

---

## [Editor Report · Decision Letter 2]

19 Apr 2024

Computerized decision support is an effective approach to select memory clinic patients for amyloid-PET

PONE-D-23-25666R2

Dear Dr. Rhodius-Meester,

We’re pleased to inform you that your manuscript has been judged scientifically suitable for publication and will be formally accepted for publication once it meets all outstanding technical requirements.

Kind regards,

Wataru Araki

Academic Editor

PLOS ONE

Additional Editor Comments (optional):

The authors have responded to all the criticisms very appropriately.
---

## [Editor Report · Acceptance letter]

8 May 2024

PONE-D-23-25666R2 

PLOS ONE

Dear Dr. Rhodius-Meester, 

I'm pleased to inform you that your manuscript has been deemed suitable for publication in PLOS ONE. Congratulations! Your manuscript is now being handed over to our production team.

Kind regards, 

on behalf of

Dr. Wataru Araki 

Academic Editor

PLOS ONE